# A New Approach Based on Glued Multi-Ultra High Molecular Weight Polyethylene Forms to Fabricate Bone Replacement Products

**DOI:** 10.3390/polym12112545

**Published:** 2020-10-30

**Authors:** Tarek Dayyoub, Aleksey Maksimkin, Fedor Senatov, Sergey Kaloshkin, Natalia Anisimova, Mikhail Kiselevskiy

**Affiliations:** 1Center for Composite Materials, National University of Science and Technology “MISIS”, 119049 Moscow, Russia; aleksey_maksimkin@mail.ru (A.M.); Senatov@misis.ru (F.S.); kaloshkin@misis.ru (S.K.); 2N.N. Blokhin National Medical Research Center of Oncology, the Ministry of Health of the Russian Federation (N.N. Blokhin NMRC of Oncology), 115478 Moscow, Russia; n_anisimova@list.ru (N.A.); kisele@inbox.ru (M.K.)

**Keywords:** ultra-high molecular weight polyethylene, glue, cellulose, thiol-ene reaction, PVA, phenol formaldehyde, mechanical properties, T-peel tests, biocompatibility

## Abstract

Three types of glue based on thiol-ene reaction, polyvinyl alcohol (PVA)/cellulose, and phenol formaldehyde were prepared and applied on modified ultra-high molecular weight polyethylene (UHMWPE) samples grafted by cellulose. In comparison with unmodified UHMWPE samples, T-peel tests on the modified and grafted UHMWPE films showed an increase in the peel strength values for the glues based on thiol-ene reaction, PVA/cellulose, and phenol formaldehyde by 40, 29, and 41 times, respectively. The maximum peel strength value of 0.62 Kg/cm was obtained for the glue based on phenol formaldehyde. Mechanical tests for the cylindrical multi-UHMWPE forms samples, made of porous UHMWPE as a trabecular layer and an armored layer (cortical layer) that consists of bulk and UHMWPE films, indicated an improvement in the mechanical properties of these samples for all glue types, as a result of the UHMWPE films existence and the increase in the number of their layers. The maximum compressive yield strength and compressive modulus values for the armored layer (bulk and six layers of the UHMWPE films using the glue based on thiol-ene reaction) were 44.1 MPa (an increase of 17%) and 1130 MPa (an increase of 36%), respectively, in comparison with one armored layer of bulk UHMWPE. A hemocompatibility test carried out on these glues clarified that the modified UHMWPE grafted by cellulose with glues based on PVA/cellulose and thiol-ene reaction were classified as biocompatible materials. These multi-UHMWPE forms composites can be considered a promising development for joint reconstruction.

## 1. Introduction

Ultra-high molecular weight polyethylene (UHMWPE), which has a molecular weight more than 10^6^ g/mol, has been a widely used polymer in recent decades. UHMWPE has good physical and chemical properties, such as high impact strength, high abrasion resistance, low coefficient of friction, excellent biocompatibility, as well as excellent chemical resistance, and it does not absorb moisture. In biomedical applications, UHMWPE has been widely used as a surface bearing material in bone replacement and total joint arthroplasty, such as artificial hips, knees, wrists, and shoulders [1,2].

For biomaterials, fabricated artificial materials should have appropriate characteristics, such as chemical compatibility and high biocompatibility, and they should also provide a structural support and form new tissue [3]. Nowadays, bone replacement products and implants particularly are made of metals, such as titanium or stainless steel. Titanium and its alloys, especially the alloy of Ti-6Al-4V, are considered the most used materials for implants. Generally, these metallic alloys should have mechanical properties close to those of the natural bones. For example, the modulus values of the first generation of orthopaedic α β titanium alloys Ti-6Al-4V, the second generation of the β Ti-alloy and SUS 316 L stainless steel are 110, 80, and 200 GPa, respectively [4]. These Young’s modulus values are considered very high in comparison with those of the natural bone, which has to be in the range of 6–30 GPa. In addition, these orthopaedic implants based on metallic alloys may have deleterious biological effects over the long term [5]. For example, titanium can cause some problems, such as corrosion, hypersensitivity, and yellow nail syndrome. Researches showed that the ions and particles of titanium can accumulate in the surrounding tissues and lead to toxic reactions in these tissues [6]. Nowadays, since the UHMWPE scaffolds do not provide the required mechanical properties for bone replacement applications, the metallic reinforcements are used in order to reinforce the UHMWPE scaffolds. On the other hand, and taking into consideration that highly oriented UHMWPE films have an elastic modulus up to 60 GPa, ultimate tensile strength up to 1.5 GPa, and elongation in the range of 3%–10% [7,8], the main idea of this work is to prepare a final bone replacement product that consists of multi-UHMWPE forms, and replacing these metallic reinforcements by highly oriented UHMWPE films, which have excellent mechanical properties without any deleterious biological effects. These bone replacement products can be fabricated by bonding some forms of UHMWPE together, such as porous, bulk, and films. Unfortunately, the main technique used to bond UHMWPE with metals or different UHMWPE forms themselves is dependent on applying high pressure and high temperature, which leads to a loss of the unique properties of the UHMWPE films. So, the best solution for the preparation of multi-UHMWPE forms composites while preserving the unique properties and characteristics for each UHMWPE form is bonding them by using glue. However, UHMWPE is an inert polymer, because it does not have any polar groups in its structure; and this means that the adhesive fixation between different UHMWPE forms is very difficult. So that, and for enhancing the adhesion properties of the UHMWPE, the surface modification of the UHMWPE can provide a good possibility to improve its adhesive fixation [9,10,11,12].

There are a lot of potential chemical products which could be used as glue for the adhesive fixation of the UHMWPE forms. Thiol-enes are a group of copolymers, which have highly ordered networks. These copolymers are used in many applications, such as biomedical and bioorganic modification applications [13,14,15,16]. The thiol-ene polymer network, which is formed in a 1:1 mode, is called “click” reaction. This reaction occurs via radical-mediated step polymerization by using UV-radiation. Thiol-ene reaction polymers provide strong bonds, and a low temperature is needed for starting the reaction in biocompatible bonding conditions [17,18]. “Click” thiol-ene chemistry involves biocompatible reactions which are used to bond substrates to biomolecules [19]. In the reference [20], they used thiol-ene click reaction on cellulose to prepare flexible hydrophilic cellulose with enhanced mechanical properties.

Polyvinyl alcohol (PVA) is a biodegradable, biocompatible, and hydrophilic synthetic polymer with excellent adhesive properties [21]. PVA has been widely applied in many applications like food, paper, textiles, wastewater treatment, and biomedical applications [22,23]. Moreover, the PVA-based scaffold has been used in tissue engineering as artificial grafts [24,25]. Cellulose (C_6_H_10_O_5_)_n_ is an organic compound. It has several advantages for medical applications, such as renewability, nontoxicity, biocompatibility, biodegradability, and excellent stability [26,27]. Cellulose can be applied in water-soluble adhesives [28]. In the reference [29], they demonstrated that the nanocomposites based on cellulose nanofiber can be successfully developed using a novel carrier system of PVA. These nanocomposites demonstrated better dispersion and good mechanical properties, and led to good compatibility between the cellulose and polyethylene matrix. 

Phenol formaldehyde (PF) resin is a synthetic polymer. It has excellent properties, such as biodegradability, good mechanical strength, heat resistance, and dimensional stability. It also has high resistance against various solvents, acids, and water. PF is a common adhesive, especially for binding wood materials in the composites industry. In the reference [30], they reported that adding cellulose to the PF matrix will increase the strength of the PF/cellulose composite, and the cellulose reinforcement will decrease the cure temperature and increase the shear strength in comparison with the neat PF adhesive.

This work aims at preparing and evaluating the mechanical properties and hemocompatibility of three potential types of glue for fabricating bone replacement products based on glued multi-UHMWPE forms, which can be considered a prospective approach for biomedical applications. As the bone consists of two structural layers formed of cortical and trabecular bone tissue, the main idea of this work is preparing a multi-UHMWPE forms bone replacement product by using a glue to bond these UHMWPE forms. The inside part of the bone replacement product is a porous UHMWPE, which is close to the porous cancellous bone tissue, and the outside part is a bulk UHMWPE, which provides compression and torsion resistance. On the other hand, this bulk UHMWPE layer is considered a soft material and does not provide the required mechanical properties in comparison with the ones of the natural bone. So, for solving this problem without using metallic alloys, an armoured layer of highly oriented and highly strengthened UHMWPE film can be glued between the porous and bulk UHMWPE layers.

## 2. Materials and Methods

### 2.1. Materials

UHMWPE GUR 4120 with a molecular weight of 5 × 10^6^ g/mol was purchased from “Ticona GmbH.” (Hesse, Germany). Benzophenone 99% was purchased from Alfa Aesar (Shanghai, China). Micro-cellulose was purchased from “Evalar Ltd.” (Altai Krai, Russia). Pentaerythritol tetrakis (3-mercaptopropionate) > 95% (PETMP) and 1,3,5-triallyl-1,3,5-triazine-2,4,6 (1H,3H,5H)-trione—98% (TATATO) were purchased from “Sigma-Aldrich” (St. Louis, MO, USA). Polyvinyl alcohol (PVA) was purchased from “Ruskhim Ltd.” (Moscow, Russia). Medical glue based on phenol formaldehyde resin.

### 2.2. Processes for Producing Cellulose Nanoparticles (CNs)

Adopting the method of Adsul et al. nano-cellulose (NC) was prepared from microcrystalline cellulose (MCC) [31,32].

### 2.3. Molding, Orientation and Drawing of UHMWPE

Depending on the approach described in the reference [8], a small amount of solvent (p-xylene) was used as a plasticizer in order to prepare UHMWPE/p-xylene gel with a ratio of 2.5 mL of solvent per 1 g of UHMWPE. This UHMWPE/p-xylene gel was fabricated at the temperature of 150 °C after storage for 20 min at this temperature, and then it was extruded by ram extruder UE-MSL (Extrusion Machinery Sales Ltd., Liversedge, UK) with a die size of 10–2 mm and an extrusion rate of 500 mm/min. The extruded gel (xerogel) was dried for 48 h at room temperature. A rolling machine BL-6175-A (Dongguan Bolon Precision Testing Machines Co., Ltd., Dongguan, China) was used to perform the first step of the orientation process for the xerogels at a temperature of 110 °C in order to obtain a draw ratio (DR) value of 1.5–2.0. Then, a special laboratory device consisting of stepping motors and sheaves system was used to pass and draw the UHMWPE film through a silicone oil bath. The oil temperature was stable within to ±0.1 °C precision. A multi-stages process of thermal orientation for the UHMWPE films was carried out stepwise at temperatures of 120 °C to DR = 5–6; 130 °C to DR = 12–15; 140 °C to DR= 20–23 and 142 °C to DR = 30–33. All processes, procedures, and conditions were broadly explained and described in the reference [33].

The mechanical properties of the UHMWPE films used to prepare the cylindrical multi- UHMWPE forms samples are shown in Table 1. The tensile measurements for the oriented UHMWPE films were carried out using Zwick/Roell Z020 universal testing machine (Zwick Roell Group, Ulm, Germany) according to ASTM D882-10 at 10 mm/min loading rate. Five samples were tested in order to determine the mechanical properties. Both ends of each film sample were glued to thin cardboards with a size of 60 mm × 50 mm before the test in order to avoid the films slipping out of the grips of the testing machine during the measurements due to the low friction coefficient values of these films [33].

### 2.4. Treatment and Modifying of the UHMWPE Films Surface

The UHMWPE films were treated by mixed acids of H_2_SO_4_ and HNO_3_ (*v/v*: 3/1) at 75 °C for one hour and then were grafted by cellulose by using a mixture of 10% nano-cellulose and 90% ethanol (*w/v*). A 5% (*w/v*) solution of benzophenone in acetone was used as a coupling agent. UV cross-linking device for UV-induced reaction at a photo density of 900 mJ/cm^2^ with an irradiation time of 20 min. All processes, procedures, and conditions were explained and described in the reference [32].

### 2.5. Preparing the Glues

#### 2.5.1. Glue Based on Thiol-ene Reaction

Stoichiometric monomers mixture was prepared by mixing PETMP/TATATO/benzophenone with a ratio of 1.1/1.0/0.1, respectively. Then, the monomers mixture was kept in a dark place for 0.5 h before the photo structuring process and gluing. The thickness of the monomer mixture, which is considered as a glue layer, was in the range of 50–100 µm. The samples which covered by the monomers mixture were put into UV cross-linking device for UV-induced reaction at a photo density of 900 mJ/cm^2^ with an irradiation time of 20 min. After that, the samples were put into oven to fully complete of the reaction at the temperature of 80 °C. The thiol-ene reaction is shown the Figure 1.

#### 2.5.2. Glue Based on PVA/Cellulose

The cellulose nanoparticles surface was modified by sulfuric acid (65%) for 10 min at the temperature of 50 °C (Figure 2a) with a ratio of 1.5 mL of the sulfuric acid for each gram of the cellulose. Then, the cellulose nanoparticles were washed and filtered by distilled water to quench the reaction and to remove the remaining acid or neutralized salt. PVA solution in distilled water was prepared by heating the solution at 80 °C for 1 h. Then, the modified cellulose nanoparticles were added to PVA solution and stirred at the temperature of 80 °C for 1 h with a ratio of the PVA/cellulose 1/0.25 (Figure 2b). The thickness of PVA/cellulose glue layer was in the range between 50–100 µm. The samples which covered by the PVA/cellulose glue layer were kept in a hot air oven for 2 h at the temperature of 80 °C to complete the reactions between PVA/cellulose glue and the cellulose on the UHMWPE films surface.

Figure 3 shows the potential reactions which can happen on the modified UHMWPE surface.

#### 2.5.3. Glue Based on Phenol Formaldehyde

The contact between the modified UHMWPE surface and the phenol formaldehyde is considered secondary interactions, such as hydrogen bonding and van der Waals forces [34]. These interactions are shown in Figure 4.

### 2.6. Preparing of Porous UHMWPE Cylindrical Samples

Porous UHMWPE samples were obtained by solid-state mixing of UHMWPE powder and NaCl and then by subsequent thermo-pressing process [35]. Mixing process was performed in a planetary ball mill Fritsch Pulverisette 5 (Fritsch GmbH, Idar-Oberstein, Germany) with low energy conditions. The ball mill was equipped with agate bowls (500 mL) and corundum grinding balls with a diameter of 10 mm to minimize penetration of impurities. The mixed powder sample was processed in 12 min cycles; each one is consisting of 10 min milling period followed by a 2 min re-cooling period. NaCl crystalline powder with a particle size ranging from 80 to 700 μm was used as a soluble material. UHMWPE and NaCl powder was taken in the ratio of 1:9 by weight.

Thermo-pressing was carried out under a load of 70 MPa and a temperature of 180 °C. UHMWPE/NaCl composite after thermo-pressing was washed with distilled water for 7 days to remove salt. The adsorbed water in the pores was removed by drying the samples at 80 °C for 3 h.

### 2.7. Preparing of Bulk UHMWPE Cylindrical Samples

Bulk UHMWPE cylindrical samples were obtained by a thermo-pressing process under a load of 40 MPa and a temperature of 190 °C. The heating time was 1 h and the storage time at 190 °C was 30 min.

### 2.8. Preparing Cylindrical Multi-UHMWPE Forms Samples

Cylindrical multi-UHMWPE forms samples were prepared by using each type of glues, as shown in Figure 5. Some layers of the UHMWPE films were used as an armored layer between the porous and the bulk layers. All types of glue were used to glue two layers of the UHMWPE films themselves and with the porous and bulk UHMWPE forms. All sample dimensions for each UHMWPE form are shown in Figure 5. In addition, in order to investigate the influence of the layers number of the UHMWPE films on the mechanical properties of the cylindrical multi-UHMWPE forms samples, samples with four and six layers of the UHMWPE films were prepared by using the glue based on thiol-ene reaction.

After preparing the cylindrical samples, they were put in the oven for 2 h at a temperature of 80 °C to get the well contact between all parts. To compare the influence of the armored UHMWPE films on the compression mechanical properties, cylindrical multi-UMWPE forms samples without adding UHMWPE films (only bulk and porous UHMWPE) were prepared for each type of glues.

### 2.9. Mechanical Testing Procedures

To evaluate the peel resistance of the adhesive bonds between flexible adherents, T-peel testing configuration according to ASTM 1876-01 was carried out on the UHMWPE films samples using Zwick/Roell Z020 universal testing machine. At least 10 measurements were applied for each type of the glues. The dimensions of the used films in this test were 10 mm × 2 mm × 0.15 mm. The thickness of glue between two glued films was 70 ± 15 µm. The load of 1 N at a constant head speed of 10 mm/min was applied. The initial peak was not considered in the peel strength values.

Since the bones of mammals are mainly subjected to mechanical stresses of compression and bending, compression tests are the preferable technique for the examination of the mechanical properties of the bone tissues (cortical or cancellous layers) in comparison with other techniques, such as tensile tests, three-point bending or torsinal tests [36]. Therefore, in order to compare the mechanical properties of the prepared composites with those of the natural bone, compression mechanical tests were carried out in this work.

Compression mechanical tests for the cylindrical multi-UHMWPE form samples were performed according to ASTM D-695 using Zwick/Roell Z020 universal testing machine. The sample dimensions are shown in Figure 5. In order to determine the compressive properties of cylindrical samples, the outside diameter of the full cylindrical multi- UHMWPE forms samples (bulk, porous and films) was used as a key parameter of the test; on the other hand, for the two armored UHMWPE layers only (bulk and films) of the cylindrical multi- UHMWPE forms samples without the inside part of the porous UHMWPE, the thickness of these two layers was used as a key parameter of the test. The samples were placed between compressive plates parallel to the surface; and then, they were compressed at a compression speed of 10 mm/min. The maximum load was recorded along with stress-strain data. In order to precisely determine the modulus and deformation values, an extensometer instrument was used. Two extensometers were connected to the upper and bottom edges of the cylindrical sample in the test zone between compressive plates during the test. Three measurements were carried out for each sample.

### 2.10. FT-IR Spectroscopy

FT-IR spectroscopy was performed by using Nicolet 380 IR-Fourier spectrometer (Thermo Fisher Scientific, Waltham, MA, USA) and an attenuated total reflection (ATR) mode: the spectral range was 4000–450 cm^−1^ with a resolution of about 0.9 cm^−1^, the accuracy of the wave number was 0.01 cm^−1^.

### 2.11. Hemocompatibility Test

Samples (disks with a diameter of 0.5 cm) were sterilized in an autoclave at 1.1 atm for 60 min. Before starting the experiment, the samples were washed with sterile 0.9% sodium chloride solution (PanEco, Moscow, Russia) and then placed one by one in the wells of a 24-well plate (Thermo Fisher Scientific, Roskilde, Denmark). Three samples of the same type were used for each test. The described study was approved by the local ethics committee of N.N. Blokhin NMRC of Oncology (Moscow, Russia).

The hemolysis was tested as described in the reference [37]. Shortly, the C57Bl/6 mouse whole blood stabilized with 60 IU/m heparin was centrifuged at 1000 rpm for 5 min to isolate red blood cells (RBCs). The RBCs were further washed 3 times with 10 mL PBS and finally suspended in 50 mL PBS. 0.5 mL RBC suspension was poured into the wells of a 24-well plate (Thermo Fisher Scientific, Waltham, MA, USA) with samples of investigated materials (UHMWPE-treated cells test), or PBS only (negative control), or 100 μL of Triton-X-100 SigmaAldrich (St. Louis, MO, USA) (positive control) and incubated at 37 °C in 5% CO_2_ atmosphere. After 2 and 4 h, these RBC suspensions were collected from wells and centrifuged at 1000 rpm for 5 min. 100 μL of the supernatants of the probes were transferred to a 96-well plate (Corning, Glendale, AZ, USA) and the optical density (OD) was measured with the MS Multiskan plate reader (Thermo Fisher Scientific, Waltham, MA USA) at 540 nm.

The percentage of hemolysis was calculated by using the Formula (1). Tested samples were described as biocompatible if the value of hemolysis was not exceeding 10% at 4 h after the start of co-incubation of the cells with the samples.
(1)$$Hemolysis\;\left(\%\right)\;=\;\frac{OD\;\left(UHMWPE-treated\;cells\;tes\right)\;-\;OD\;\left(Negative\;Control\right)}{OD\;\left(Positive\;Control\right)\;-\;OD\;\left(Negative\;Control\right)}\;\times 100$$

Lactate dehydrogenase (LDH) release was evaluated to assess the cell membrane integrity with the Pierce LDH Cytotoxicity Assay Kit (Thermo Fisher Scientific, Waltham, MA, USA) in accordance with the instruction of the manufacturer. Mononuclear leucocytes (ML) were separated from mouse whole blood stabilized with 60 IU/m heparin. 1 mL blood of C57Bl/6 mice (m = 20 ± 1 g) diluted with an equal amount of 1X PBS. Then gently overlaid the Ficoll (PanEco, Moscow, Russia) with diluted blood was centrifuged at 1700 rpm for 20 min at room temperature. Later the ML layer was collected carefully from the plasma/Ficoll interface and washed twice with RPMI-1640 medium (PanEco, Moscow, Russia) at 1000 rpm for 5 min. ML was suspended in a complete growth medium based on RPMI-1640 medium containing 10% of fetal bovine serum (HyClone, Thermo Fisher Scientific, Waltham, MA, USA), 4 mM L-glutamine and 1% penicillin/streptomycin (both from PanEco, Moscow, Russia). The resultant ML suspension with a concentration of 5.2 × 10^4^ of the cells was incubated with investigated samples (UHMWPE-treated cell tests) at 37 °C in 5% CO_2_ atmosphere for 24 h. Untreated cells incubated at the same conditions were used as Control. At the end of the incubation, the plates were gently shaken to ensure that the LDH was evenly distributed in the medium. Quantitative analysis was performed on the cell culture supernatant (50 μL/well) after centrifugation at 1000 rpm for 5 min. OD in control and UHMWPE-treated cell tests were measured using a MS Multiskan plate reader (Labsystem, Thermo Fisher Scientific, Waltham, MA, USA) at 492 nm opposite 690 nm in accordance with formula (2). The samples will be considered as biocompatible material if the LDH release (UHMWPE -treated cell tests) will not significantly differ from the LDH release (control).
(2)LDH Release = OD(492 nm) − OD(690 nm)

All experiments were performed with triplicate assays and repeated three times using separated experiments. OD measurements were presented as mean ± standard deviation (SD). A comparative analysis was performed using the t- test. The difference was considered significant at *p* < 0.05.

## 3. Results and Discussion

Table 2 shows the T-peel results for the glued UHMWPE films before and after treatment by cellulose grafting measured according to ASTM 1876-01. The results are the average values of the peel strength after the initial peak. As it can be seen in Table 2, the mechanical properties were improved after the treatment by cellulose grafting for all types of glue. In addition, the peel strength values were increased for the glue based on the thiol-ene reaction, the glue based on PVA/cellulose, and the glue based on phenol formaldehyde by 40, 29, and 41 times, respectively. The maximum peel strength value was obtained for the glue based on phenol formaldehyde, and it was 0.62 Kg/cm.

Table 3 shows the mechanical test results for the cylindrical multi-UHMWPE forms samples without and with two layers of the UHMWPE films according to ASTM D-695. As it can be seen in Table 3, the existence of the UHMWPE films was led to improve the mechanical properties of the multi-UHMWPE forms samples for all glue types. The best mechanical properties were obtained for the glue based on the thiol-ene reaction in comparison with the other ones, where the maximum compressive yield strength value was 12.9 MPa and the maximum Young’s modulus value was 0.862 GPa. In addition, as can be seen in Table 3 for the full cylindrical multi-UHMWPE forms samples with the inside part of porous UHMWPE, the Young’s modulus was increased for the glues based on the thiol-ene reaction, PVA/cellulose, and phenol formaldehyde by 26%, 17%, and 0.4%, respectively.

Table 4 shows the mechanical test results for the only two armored layers (bulk and 2 layers of films) of the cylindrical multi-UHMWPE forms samples without the inside part of the porous UHMWPE according to ASTM D-695. As can be seen in Table 4, the addition of the UHMWPE films led to an increase in the compressive yield strength value by about 13% for the glue based on thiol-ene reaction, by about 19% for the glue based on PVA/Cellulose, and by about 3% for the glue based on phenol formaldehyde. Moreover, the addition of the UHMWPE films was led to an increase in the Young’s modulus for the glues based on thiol-ene reaction, PVA/cellulose and phenol formaldehyde by 20%, 26%, and 2%, respectively. For the two UHMWPE armored layers (bulk and two layers of films), the maximum compressive yield strength value was obtained for the glue based on thiol-ene, and it was 41.8 MPa, and the maximum Young’s modulus value was obtained for the glue based PVA/cellulose, and it was 1050 MPa.

In order to investigate the influence of the layers number of the UHMWPE films on the mechanical properties of the cylindrical multi-UHMWPE forms samples, samples with four and six layers of the UHMWPE films were prepared by using glue based on thiol-ene reaction. By taking into consideration the fact that the maximum compressive yield strength of the cylindrical samples with two films layers was obtained for the glue based on thiol-ene reaction, this type of glue was used to prepare these samples. Table 5 shows the mechanical test results for these samples according to ASTM D-695. As it can be seen in Table 5, the increase in the number of the layers of the UHMWPE films was led to improve the mechanical properties of the multi-UHMWPE forms samples. The maximum compressive yield strength value for the full cylindrical UHMWPE (porous, bulk and films) was 13.8 MPa when the layers number of the UHMWPE films was six layers. In addition, the mechanical test results for the two armored layers (bulk and the layers of films) of the cylindrical multi-UHMWPE forms samples based on thiol-ene glue without the inside part of porous UHMWPE and depending on the number of the layers of the UHMWPE films, it was shown that the increase of the layers of the UHMWPE films led to an increase in the compressive yield strength value, in comparison with bulk armored layer without the layers of films, by 16% for four layers of the UHMWPE films, and by 17% for six layers of the UHMWPE films. Moreover, the increase of the layers of the UHMWPE films led to an increase in the Young’s modulus value, in comparison with bulk armored layer without the layers of films, by 35% for four layers of the UHMWPE films, and by 36% for six layers of the UHMWPE films. The maximum compressive yield strength and Young’s modulus values were obtained for the six layers of the UHMWPE films, and they were 44.1 MPa and 1130 MPa, respectively.

Figure 6 shows the FT-IR spectra of the virgin UHMWPE film, modified UHMWPE film by cellulose grafting and modified UHMWPE films glued by the three types of glue. The FT-IR spectrum peaks for all components are shown in Table 6. From Table 6 and Figure 6, it can be confirmed that the contact between the modified UHMWPE surface by cellulose and the phenol formaldehyde is considered secondary interactions, such as hydrogen bonding and van der Waals forces [34]. Moreover, for the glue based on PVA/cellulose, the peak at 1172 cm^-1^ could be related to the reaction between sulphuric group of the PVA/cellulose glue and hydroxyl group of the grafted cellulose on the modified UHMWPE surface (Figure 3a). In addition, for the glue based on the thiol-ene reaction, the peaks at 1688 cm^−1^, 1733 cm^−1^, and 3460 cm^−1^ could be related to the reaction between sulphide group of the thiol and carbonyl groups of the grafted cellulose on the modified UHMWPE surface.

Table 7 shows the induced hemolysis results for the treated UHMWPE films grafted by cellulose, and for the modified UHMWPE films, which glued by the glues based on PVA/Cellulose and thiol-ene reaction. As it can be seen in Table 7, the percentage of hemolysis induced by the studied materials during 4 h of the co-incubation with RBC did not significantly exceed 10%. It should be noted that the PVA/cellulose glue demonstrated a minimal cell damaging effect (4% ± 1.5%), whereas the hemolytic activity of the thiol-ene reaction glue reached up to 9% ± 3.3 %.The LDH released from UHMWPE-treated WBCs did not differ significantly from the control with untreated cells (*p* > 0.05). So, the modified UHMWPE by cellulose, and also samples glued by PVA/cellulose and the thiol-ene reaction are classified as biocompatible materials.

Since the obtained mechanical results revealed the good properties of the multi-UHMWPE forms composites (Figure 7, Figure 8 and Figure 9) glued by some glue types based on thiol-ene reaction, PVA/Cellulose and phenol formaldehyde; and also since these materials can be considered biocompatible; they can be considered as a promising development for joint reconstruction, such as implants for the replacement of bone defects or screws for osteofixation and orthognathic surgery. In addition, the Young’s modulus value (compressive modulus) of these composites is close to that of the natural bone, which is in range of 6–30 GPa [45,46], in comparison with the commercially available products based on metals, which has a very high modulus value as presented in Section 1 [4]. This will lead to a successful bone remodeling and prevent bone resorption. Of course, the obtained compression mechanical properties of the prepared multi-UHMWPE forms composites do not fully match ones of the natural bone, but the proposed approach of using highly strengthened UHMWPE films as a reinforcement for the implants based only on UHMWPE without metallic reinforcement can be considered a promising approach.

## 4. Conclusions

Three types of glue based on thiol-ene reaction, PVA/Cellulose and phenol formaldehyde were prepared and applied on modified UHMWPE films grafted by cellulose. T-peel tests indicated to the improvement of the mechanical properties of the modified UHMWPE films for all types of glue after being treated by cellulose grafting. The peel strength value increased for the glues based on thiol-ene reaction, PVA/Cellulose and phenol formaldehyde by 40, 29, and 41 times, respectively. The maximum peel strength value of 0.62 Kg/cm was obtained for the glue based on phenol formaldehyde.

Compression mechanical tests for the cylindrical multi-UHMWPE forms samples showed that the existence of the UHMWPE films led to the mechanical properties improvement of the cylindrical multi-UHMWPE forms samples for all glue types. In addition, it should be noted that the mechanical properties of the multi-UHMWPE forms samples improved due to the increase in the number of the UHMWPE films layers. For the full multi-UHMWPE forms (porous, bulk, and six layers of films) using the glue based on thiol-ene, the maximum compressive yield strength value was 12.9 MPa and compressive modulus value was 862 MPa. For the two UHMWPE armored layers (bulk and six layers of the UHMWPE films) and by using the glue based on thiol-ene reaction, the maximum compressive yield strength and Young’s modulus values were 44.1 MPa (an increase of 17%) and 1130 MPa (an increase of 36%), respectively, in comparison with one armored layer of bulk UHMWPE.

The FT-IR spectra results provided that the contacts among the treated surface of UHMWPE films grafted by cellulose and all types of glue were generally secondary interactions.

The hemocompatibility studies showed that the investigated samples of the treated UHMWPE films grafted by cellulose, the samples glued by the glue based on PVA/cellulose and samples glued by the glue based on thiol-ene reaction can be classified as biocompatible materials, since the induced hemolysis did not exceed 10%; and also, there were no signs for damage in the WBC membranes during co-incubation.

Bone replacement products based on glued multi-UHMWPE forms are considered a modern approach, which is important for medical applications. By using this approach, it is possible to prepare cheap bone replacement products. The challenging and important problems for future studies are the enhancement of the glued fixation and the improvement of the mechanical properties of the prepared composites using the proposed technique that depends on the oriented UHMWPE films as a reinforcement material in order to fully match the mechanical properties of the natural bone.

## Figures and Tables

**Figure 1 polymers-12-02545-f001:**
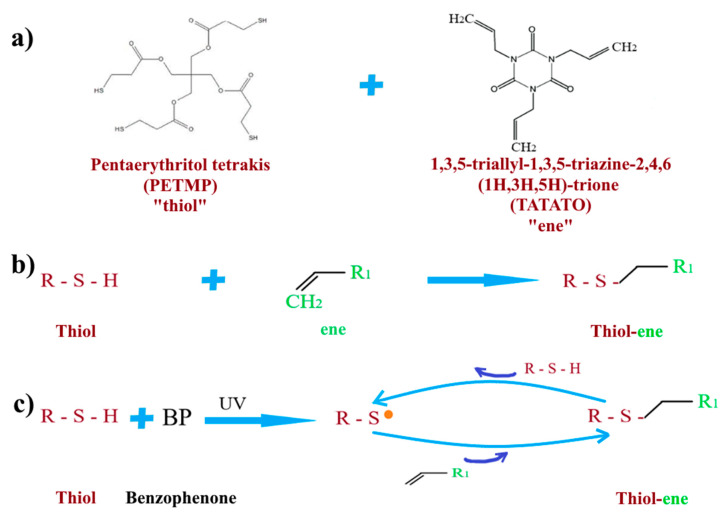
Thiol-ene reaction. (**a**) molecular formulas of PETMP and TATATO; (**b**) and (**c**) mechanism of thiol-ene reaction.

**Figure 2 polymers-12-02545-f002:**
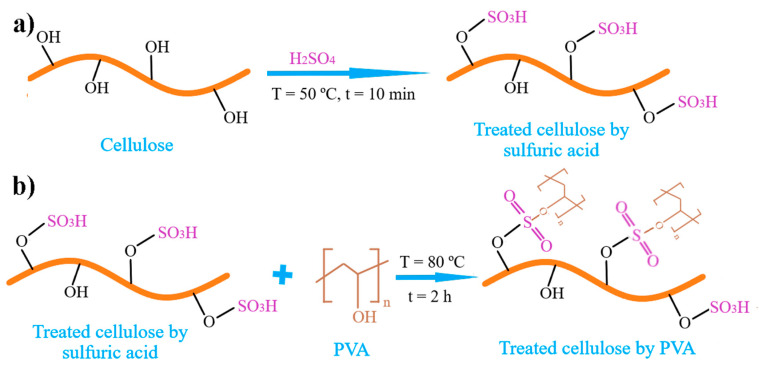
The potential reactions of the glue based on PVA/Cellulose: (**a**) the modifying reaction of cellulose nanoparticles surface by sulfuric acid; (**b**) the reaction between PVA and modified cellulose nanoparticles.

**Figure 3 polymers-12-02545-f003:**
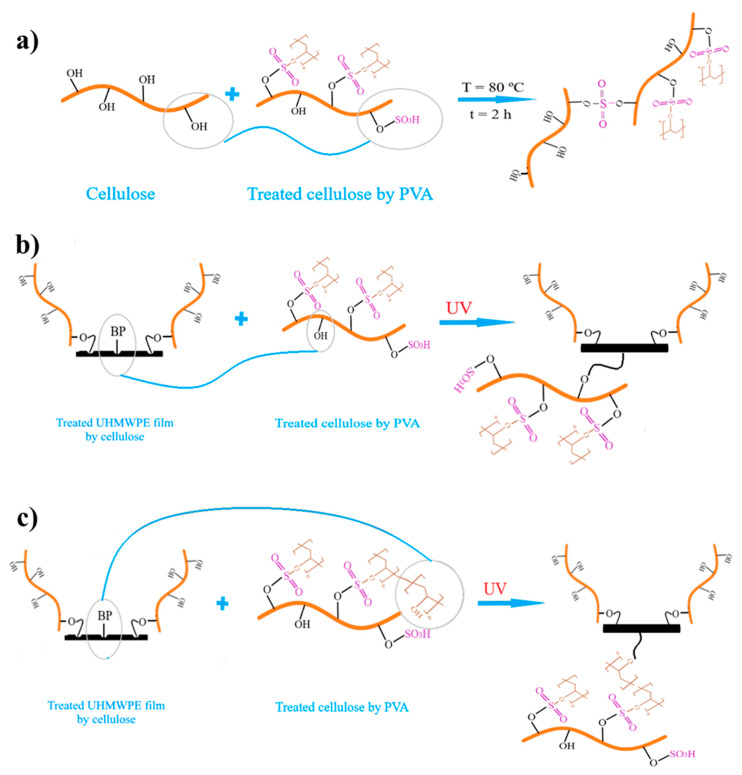
The potential reactions between the modified UHMWPE surface and PVA/Cellulose glue: (**a**) hydroxyl groups of cellulose and sulfuric acid groups, (**b**) residual benzophenone and hydroxyl groups of the cellulose of the glue, (**c**) residual benzophenone and hydroxyl groups of the PVA of the glue.

**Figure 4 polymers-12-02545-f004:**
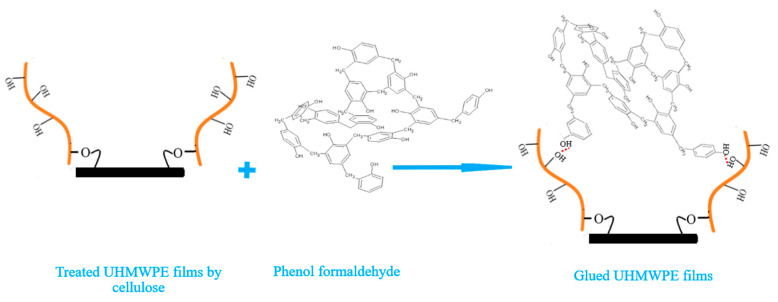
Hydrogen bonds between cellulose and phenol formaldehyde.

**Figure 5 polymers-12-02545-f005:**
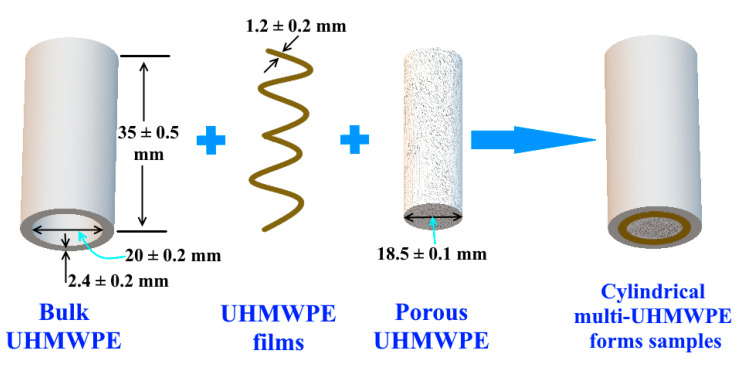
Cylindrical multi-UHMWPE forms samples, which consists of 3 UHMWPE forms. The armored layer consists of bulk and films of UHMWPE; and the trabecular layer consists of porous UHMWPE.

**Figure 6 polymers-12-02545-f006:**
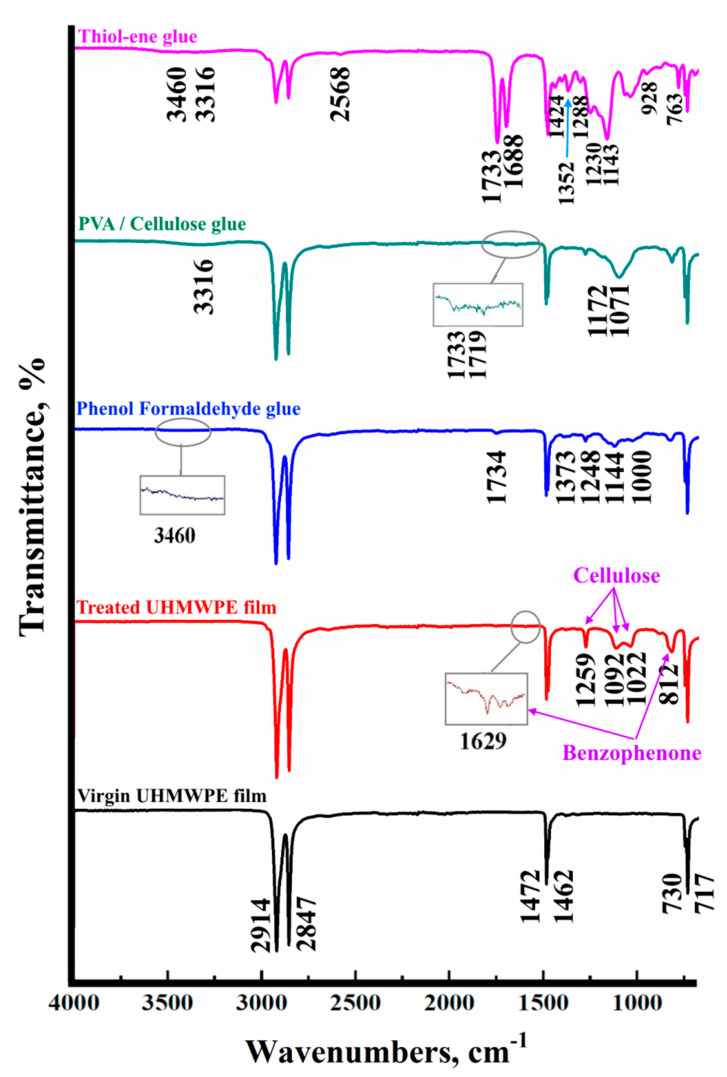
The FT-IR spectra of the virgin UHMWPE film, treated UHMWPE film by cellulose grafting and treated UHMWPE films glued by glues based on thiol-ene reaction, PVA/Cellulose and phenol formaldehyde.

**Figure 7 polymers-12-02545-f007:**
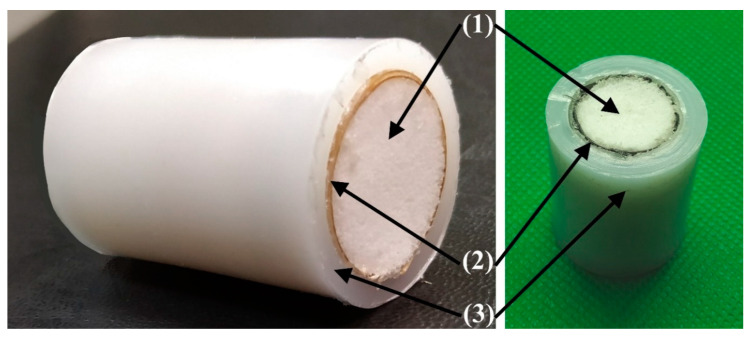
Multi-UHMWPE forms composite of porous UHMWPE (**1**), armored layer of highly oriented and strengthened UHMWPE films (**2**) and bulk UHMWPE (**3**).

**Figure 8 polymers-12-02545-f008:**
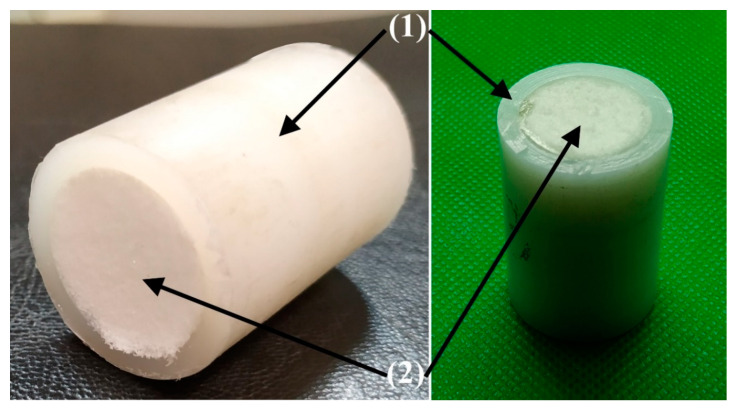
Multi-UHMWPE forms composite of bulk (**1**) and porous (**2**) UHMWPE.

**Figure 9 polymers-12-02545-f009:**
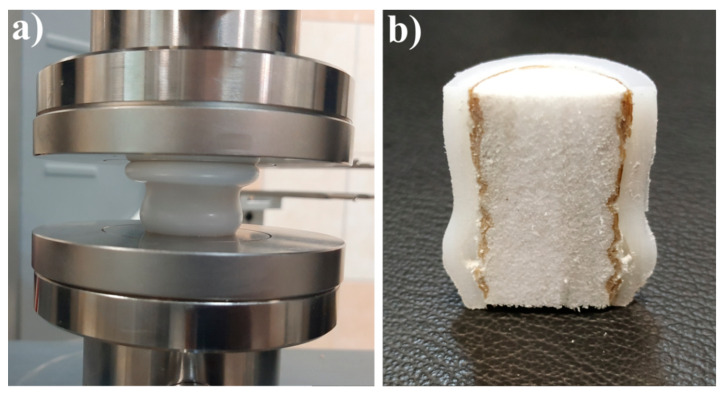
Cylindrical multi- UHMWPE forms samples, made of porous UHMWPE as trabecular layer and armored layer (cortical layer) that consists of bulk and films of UHMWPE. (**a**) The sample appearance in the end of the mechanical test; (**b**) dissected sample that shows the fixation of the glued multi- UHMWPE forms after the mechanical test.

**Table 1 polymers-12-02545-t001:** The mechanical properties of the UHMWPE films used to prepare the cylindrical multi-UHMWPE forms samples (± standard deviation).

Tensile Strength, MPa	Young’s Modulus, GPa	Elongation, %
1100 ± 50	50 ± 3	3.4 ± 1.0

**Table 2 polymers-12-02545-t002:** T-peel results according to ASTM 1876-01 (± standard deviation).

Type of Glue	Before the Treatment	After the Treatment
	Peel Strength, Kg/cm	Peel Strength, Kg/cm
Glue based on thiol-ene reaction	0.010 ± 0.002	0.40 ± 0.08
Glue based on PVA/Cellulose	0.015 ± 0.002	0.43 ± 0.09
Glue based on phenol formaldehyde	0.015 ± 0.002	0.62 ± 0.10

**Table 3 polymers-12-02545-t003:** Mechanical test results for the cylindrical multi-UHMWPE forms samples according to ASTM D-695.

Type of Glue	Cylindrical Multi-UHMWPE Forms Samples (Bulk and Porous)			Cylindrical Multi-UHMWPE Forms Samples (Bulk, Porous and 2 Layers of Films)		
	Compressive Modulus, GPa	Compressive Yield Strength, MPa	Deformation at Compressive Yield Strength, %	Compressive Modulus, GPa	Compressive Yield Strength, Mpa	Deformation at Compressive Yield Strength, %
Glue based on thiol-ene reaction	0.642 ± 0.007	11.9 ± 1.1	19.2 ± 0.6	0.862 ± 0.009	12.9 ± 1.0	17.7 ± 0.9
Glue based on PVA/Cellulose	0.601 ± 0.008	9.87 ± 0.9	13.8 ± 0.8	0.720 ± 0.010	12.4 ± 0.9	18.5 ± 0.5
Glue based on phenol formaldehyde	0.590 ± 0.006	10.1 ± 1.0	14.2 ± 0.9	0.592 ± 0.011	10.5 ± 0.7	15.5 ± 0.6

**Table 4 polymers-12-02545-t004:** Mechanical test results for the armored layers of the cylindrical multi- UHMWPE forms samples according to ASTM D-695.

Type of Glue	Bulk Armored Layer			Bulk Armored Layer with 2 Layers of Films		
	Compressive Modulus, GPa	Compressive Yield Strength, MPa	Deformation at Yield Compressive Strength, %	Compressive Modulus, GPa	Compressive Yield Strength, MPa	Deformation at Compressive Yield Strength, %
Glue based on thiol-ene reaction	0.728 ± 0.007	36.5 ± 0.9	19.1 ± 1.1	0.912 ± 0.008	41.8 ± 0.5	17.2 ± 1.1
Glue based on PVA/Cellulose	0.775 ± 0.011	29.9 ± 1.2	20.0 ± 0.8	1.050 ± 0.006	37.1 ± 0.9	16.2 ± 0.9
Glue based on phenol formaldehyde	0.677 ± 0.009	30.8 ± 0.8	18.2 ± 0.9	0.693 ± 0.009	33.4 ± 0.8	13.1 ± 1.3

**Table 5 polymers-12-02545-t005:** Mechanical test results for the cylindrical multi- UHMWPE forms samples glued by thiol-ene glue with four and six layers of the UHMWPE films according to ASTM D-695.

Type of Sample	Four Layers of the UHMWPE Films			Six Layers of the UHMWPE Films		
	Compressive Modulus, GPa	Compressive Yield Strength, MPa	Deformation at Compressive Yield Strength, %	Compressive Modulus, GPa	Compressive Yield Strength, MPa	Deformation at Compressive Yield Strength, %
Cylindrical multi-UHMWPE forms samples (bulk, porous and films)	0.836 ± 0.016	13.3 ± 0.9	19.5 ± 0.6	0.884 ± 0.011	13.8 ± 1.5	17.8 ± 0.5
Cylindrical multi-UHMWPE forms samples (two armored layers (bulk and films)	1.120 ± 0.011	43.7 ± 1.5	18.1 ± 0.6	1.130 ± 0.010	44.1 ± 1.1	17.8 ± 0.2

**Table 6 polymers-12-02545-t006:** The FT-IR spectrum peaks for all components.

UHMWPE [38,39]		Benzophenone [40]		Cellulose [41,42]	
Wave Number, cm^−1^	Functional Group	Wave Number, cm^−1^	Functional Group	Wave Number, cm^−1^	Functional Group
717	rocking vibration peak due to the high degree of polymerization and long molecular chain of UHMWPE	812	C–CO–C sym. str.	1022	C–O stretching group of 3,6-anhydrogalactose
730					
				1092	
1462	in-plane bending vibration peak of C–H				
1472		1629	C=O stretch	1259	C–OH bending at C_6_
2847	sym. stretching vibration peak of C–H				
2914	asym. stretching vibration peak of C–H				
**Phenol formaldehyde** [43,44]		**PVA/Cellulose** [30,43]		**Thiol-ene** [43]	
Wave number, cm^−1^	Functional group	Wave number, cm^−1^	Functional group	Wave number, cm^−1^	Functional group
1019	alcohol C–O stretching vibrations	1071	stretching vibrations of the S=O group (PVA) bending vibration in cellulose/PVA	763	str vib –C–S
				928	CH_3_–S (CH_3_ rocking vib. Sal. Compounds)
1144	C–O stretch (methylol)	1172	=SO_2_ asymmetric and symmetric stretching Vibrations dialkyl sulphites, (RO)_2_SO	1143	C–O stretch (methylol)
				1230	–CH_2_–S– (CH_2_ def vib)
1248	C–O–C stretch	1248	C–O–C stretch	1288	
				1352	sym-triazines (Ring str, at least one band)
1373	OH in plane (phenolic)	1719	C=O stretch (overlapped with OH scissors of water)	1389	
				1424	CH_3_CH_2_–S– (CH_2_ def vib)
1734	C=O stretch (overlapped with OH scissors of water)	1733		1688	dialkyl thiolesters, R–CO–SR (C=O str)
				2568	str vib S–H
3460	Ortho-substituted (X–C–O X=OR)	3316	vOH cellulose/PVA	3316	vOH
				3460	Ortho-substituted (X–C–OX=OR)

**Table 7 polymers-12-02545-t007:** The hemocompatibility test results (± standard deviation).

Sample	Hemolysis, %		LDH Release
	Incubation Time, h		
	2	4	
Treated UHMWPE films grafted by Cellulose	3 ± 0.1	5 ± 1.0	1805 ± 0.039 (*p* = 0.093)
Glue based on thiol-ene reaction	3 ± 0.2	4 ± 1.5	1814 ± 0.096 (*p* = 0.216)
Glue based on PVA/Cellulose	4 ± 0.1	9 ± 3.3	2033 ± 0.051 (*p* = 0.071)
Untreated cells (Control)	-	-	1933 ± 0.005

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
