# Peer review of "A New Approach Based on Glued Multi-Ultra High Molecular Weight Polyethylene Forms to Fabricate Bone Replacement Products"

_polymers, 2020, doi:10.3390/polym12112545_

Round 1
Reviewer 1 Report
In the submitted paper, the authors investigated the preparation of three types of glue using thiol-ene reaction, PVA/cellulose and phenol formaldehyde. The mechanical properties of the resulting polymers were examined. The following issues should be addressed. 1. The major concern is that the authors claimed that the synthesized materials could be potentially used for bone replacement. However, no data could support this claim, and at least, the biocompatibility of the synthesized materials should be investigated. 2. Could the authors compare the results between the synthesized glue and commercially available products for bone replacement? Among the three synthesized glue, which one is the most promising for bone replacement?Author Response
- The major concern is that the authors claimed that the synthesized materials could be potentially used for bone replacement. However, no data could support this claim, and at least, the biocompatibility of the synthesized materials should be investigated.
Thank you for your comment, but the biocompatibility of the synthesized materials was investigated in our manuscript. Hemocompatibility is one of the major criteria, which determine the clinical applicability of blood-contacting biomaterials. Hemocompatibility is a measure of the thrombotic response induced by a material or device in contact with blood that will lead to the activation of the blood coagulation cascade, including platelet response, complement activation, and coagulation cascade initiation. Please, take into your consideration that all conditions and procedures of this test were explained in our manuscript; and depending on the results presented in Table 7, the treated UHMWPE films grafted by cellulose and the prepared samples glued by both PVA/Cellulose and thiol-ene glues were classified as biocompatible materials.
Our previous works [reference № 37 in this article; Fedor Senatov, Gulbanu Amanbek, Polina Orlova, Mikhail Bartov, Tatyana Grunina, Evgeniy Kolesnikov, Aleksey Maksimkin et al. Biomimetic UHMWPE/HA scaffolds with rhBMP-2 and erythropoietin for reconstructive surgery // Materials Science & Engineering C 111 (2020) 110750] showed that porous UHMWPE scaffolds are biocompatible, and they do not cause any cytotoxicity or hemolysis. In vivo studies for the bone implantation of porous scaffolds carried on mice demonstrated the interactions material/tissue, provisional matrix formation, granulation tissue development, fibrosis and fibrous capsule development [37]. No negative tissue reaction on the implant after implantation in vivo was observed [37]. It was shown in reference [Fedor Senatov, Gulbanu Amanbek, Polina Orlova, Mikhail Bartov, Tatyana Grunina, Evgeniy Kolesnikov, Aleksey Maksimkin et al. Biomimetic UHMWPE/HA scaffolds with rhBMP-2 and erythropoietin for reconstructive surgery // Materials Science & Engineering C 111 (2020) 110750] that the porous UHMWPE scaffold with hydroxyapatite induced the growth of new bone tissue. So, in this work the hemocompatibility test were carried out in order to investigate the biocompatibility of the treated UHMWPE grafted by cellulose, and the glues based on thiol-ene and PVA/Cellulose.
On the other hand, and since the mechanical properties are one of the main factors of the potential synthesized materials for bone replacement, all mechanical properties of the prepared samples were provided in our manuscript.
- Could the authors compare the results between the synthesized glue and commercially available products for bone replacement?
Please, be considered that in the introduction section, a brief of commercially available products for bone replacement, which are particularly made of metals, such as titanium and stainless steel, was provided. Since the important factor for these products is the mechanical properties, especially the modulus value, these commercially products based on metals have very high modulus values in comparison with those of the natural bone. Moreover, these products based on metals have deleterious biological effects, and these biological effects are absent for the UHMWPE. On the other hand, commercially products for bone replacement based only on UHMWPE are not available in the market for better comparison. Our approach by using glues and an armored layer of UHMWPE films is considered a new one.
Please, check the following reformulated paragraph:
“Since the obtained mechanical results revealed the good properties of the multi-UHMWPE forms composites (Figure 7 and Figure 8) glued by some glue types based on thiol-ene reaction, PVA/Cellulose and phenol formaldehyde; and also since these materials can be considered biocompatible; they can be considered as a promising development for joint reconstruction, such as implants for the replacement of bone defects or screws for osteofixation and orthognathic surgery. In addition, the Young’s modulus value (compressive modulus) of these composites is close to the one of the natural bone, which is in range of 6-30 GPa [45, 46], in comparison with the commercially available products based on metals, which has very high modulus value as presented in the introduction section [4]. This will lead to a successful bone remodeling and prevent bone resorption. Of course, the obtained compression mechanical properties of the prepared multi-UHMWPE forms composites do not fully match ones of the natural bone, but the proposed approach of using highly strengthened UHMWPE films as a reinforcement for the implants based only on UHMWPE without metallic reinforcement can be considered a promising approach.”
- Among the three synthesized glue, which one is the most promising for bone replacement?
Please take into your consideration that all these glues are promising for bone replacement. Here, some factors depending on the mechanical properties, biocompatibility, availability, price and process-ability should be considered in order to determine the most promising one. In the point view of the mechanical properties, we used the glue based thiol-ene reaction in order to investigate the influence of the layers number of the UHMWPE films on the mechanical properties of the cylindrical multi-UHMWPE forms samples, because this glue has the the maximum compressive yield strength in comparison with the others. On the other hand, in the point view of the biocompatibility, the glue based on PVA/Cellulose is considered a little bit more biocompatible material than the glue based on thiol-ene reaction. Moreover, the cheapest glue is the glue based on phenol formaldehyde, which can be purchased in the pharmacy, but its mechanical properties were inferior to those of others.
Reviewer 2 Report
Title: A new approach based on glued multi-UHMWPE forms to fabricate bone replacement products
Authors: T. Dayyoub and colleagues
Overall assessment:
I think that the topic adopted by the authors may be interesting and important for this literature. However, there are many irrelevancies in the manuscript, which may be due to the lack of careful and objective reading. In the present status, many readers cannot appreciate the essence of the study at all. Therefore, the authors must revise the manuscript thoroughly while reducing the irrelevancies.
Followings are the issues that the authors must overcome for continuing the reviewing process:
Specific comments:
- I recommend the authors to ask whether the acronyms such as UHWPE and PVA can be used for the title and abstract to the editorial office.
- There are no descriptions on the methods how to determine the properties of UHMWPE films listed in Table 1. If these values were obtained from some actual tests, brief descriptions must be addressed. Otherwise, the source of these values must be listed in the reference section.
- “Properties of UHMWPE film” and “UHMWPE film” are excessive and should be removed from Table 1.
- In Fig. 5, unit is missing.
- The caption of Fig. 5 is too terse. More information is necessary.
- For the right figure in Fig. 5, the explanation is addressed as “Cylindrical UHMWPE multi-forms samples”. However, in the caption, the corresponding description is “Cylindrical multi- UHMWPE forms samples”. I think the orders of the words should be unified.
- When reading the manuscript alone, it is very difficult to understand what the portion in Fig. 5 is used for T-peel tests.
- The descriptions on the mechanical tests are extremely terse. Even if the mechanical tests were conducted according to the standards listed here, the descriptions must be more elaborated. For example, it is ambiguous why the authors conducted the compression tests. It is necessary to demonstrate the properties obtained from the mechanical tests in this subsection.
- The equations must be described using some software such as MathType or the function incorporated to the MSWord. In fact, Eqs. (1) and (2) are blurred.
- In Table 3, the values of Young’s modulus are listed as those of “Compressive modulus”. However, there is a concern that these values are not confidential.
When reading the subsection 2.9, there are no descriptions how to measure the strain during the compression test. If the strain was measured from the crosshead movement, it could not be obtained accurately because of the take up of play in the testing machine and eccentricity in loading.
At least, the descriptions on the method how to measure the strain must be denoted in the subsection 2.9. If the authors think that the compression testing data are confidential enough, they must denote the issue clearly in the manuscript.
- The presentation of Table 6 must be elaborated.
- It is difficult to understand why the authors use a pair of photographs for Fig. 7. I think that either photograph is not necessary. This issue is also applicable to Fig. 8.
- The caption of Fig. 9 is too terse. It is necessary to demonstrate what the authors would like to represent by using the left and right photographs in the caption. This issue is also applicable to Fig. 10.
- The authors insist on the feasibility of the materials for the replacement of bone defects, etc., because the Young’s modulus value of the materials is close to that of bone. However, this issue is dubious at all because of the irrelevancies in measuring the Young’s modulus. Additionally, there are no descriptions on the Young’s modulus values of bones; therefore, it is difficult to evaluate the quality of the materials fabricated in this study.
Recommendation
Thorough revisions are required and the authors must provide a point-by-point response to the comments while preparing the revised version. Several descriptions are too terse to understand the issues experimentally conducted precisely. In contrast, the arrangement of the manuscript is not adequate; therefore, it is often very tedious to read. Additionally, there are huge ambiguities as described in the specific comments. Although I’d like to read the revised version, I must provide a negative evaluation if the revisions are not adequately conducted. I’d like to receive the revised version in due course.
Author Response
1. I recommend the authors to ask whether the acronyms such as UHWPE and PVA can be used for the title and abstract to the editorial office.
Please, be informed that the acronyms in the title and abstract were changed.
2. There are no descriptions on the methods how to determine the properties of UHMWPE films listed in Table 1. If these values were obtained from some actual tests, brief descriptions must be addressed. Otherwise, the source of these values must be listed in the reference section.
Would you please check the following reformulated paragraph:
“The mechanical properties of the UHMWPE films used to prepare the cylindrical multi- UHMWPE forms samples are shown in Table 1. The tensile measurements for the oriented UHMWPE films were carried out using Zwick/Roell Z020 universal testing machine according to ASTM D882-10 at 10 mm/min loading rate. Five samples were tested in order to determine the mechanical properties. Both ends of each film sample were glued to thin cardboards with a size of 60 mm × 50 mm before the test in order to avoid the films slip out of the grips of the testing machine during the measurements due to the low friction coefficient values of these films [33].”
3. “Properties of UHMWPE film” and “UHMWPE film” are excessive and should be removed from Table 1.
Would you please check the following reformulated Table 1:
Table 1. The mechanical properties of the UHMWPE films used to prepare the cylindrical multi- UHMWPE forms samples (± standard deviation).
|
Tensile strength, MPa |
Young’s modulus, GPa |
Elongation, % |
|
1100 ± 50 |
50 ± 3 |
3.4 ± 1.0 |
4. In Fig. 5, unit is missing.
The dimensions unit was added to Figure 5.
5. The caption of Fig. 5 is too terse. More information is necessary.
The caption of Figure 5 was changed as follows:
Figure 5. Cylindrical multi- UHMWPE forms samples, which consists of 3 UHMWPE forms. The armored layer consists of bulk and films of UHMWPE; and the trabecular layer consists of porous UHMWPE.
6. For the right figure in Fig. 5, the explanation is addressed as “Cylindrical UHMWPE multi-forms samples”. However, in the caption, the corresponding description is “Cylindrical multi- UHMWPE forms samples”. I think the orders of the words should be unified.
The term of “Cylindrical multi- UHMWPE forms samples” was used in Figure and in its caption.
7. When reading the manuscript alone, it is very difficult to understand what the portion in Fig. 5 is used for T-peel tests.
Please, take into your consideration that the T-peel tests were carried out for the UHMWPE films only. These tests cannot be carried out for the cylindrical multi- UHMWPE forms samples.
Please, check the reformulated paragraph:
“To evaluate the peel resistance of the adhesive bonds between flexible adherents, T-peel testing configuration according to ASTM 1876-01 was carried out on the UHMWPE films samples. At least 10 measurements were applied for each type of the glues. The dimensions of the used films in this test were 10 mm × 2 mm × 0.15 mm. The thickness of glue between two glued films was 70 ± 15 µm. The load of 1 N at a constant head speed of 10 mm/min was applied. The initial peak was not considered in the peel strength values.”
8. The descriptions on the mechanical tests are extremely terse. Even if the mechanical tests were conducted according to the standards listed here, the descriptions must be more elaborated. For example, it is ambiguous why the authors conducted the compression tests. It is necessary to demonstrate the properties obtained from the mechanical tests in this subsection.
Please, check the reformulated paragraphs:
“Since the bones of mammals are mainly subjected to mechanical stresses of compression and bending, compression tests are the preferable technique for the examination of the mechanical properties of the bone tissues (cortical or cancellous layers) in comparison with other techniques, such as tensile tests, three-point bending or torsinal tests [36]. Therefore, and in order to compare the mechanical properties of the prepared composites with those of the natural bone, compression mechanical tests were carried out in this work.
Compression mechanical tests for the cylindrical multi- UHMWPE form samples were performed according to ASTM D-695. The sample dimensions are shown in Figure 5. In order to determine the compressive properties of cylindrical samples, the outside diameter of the full cylindrical multi- UHMWPE forms samples (bulk, porous and films) was used as a key parameter of the test; on the other hand, for the two armored UHMWPE layers only (bulk and films) of the cylindrical multi-UHMWPE forms samples without the inside part of the porous UHMWPE, the thickness of these two layers was used as a key parameter of the test. The samples were placed between compressive plates parallel to the surface; and then, they were compressed at a compression speed of 10 mm/min. The maximum load was recorded along with stress-strain data. In order to precisely determine the modulus and deformation values, an extensometer instrument was used. Two extensometers were connected to the upper and bottom edges of the cylindrical sample in the test zone between compressive plates during the test. Three measurements were carried out for each sample.”
9. The equations must be described using some software such as MathType or the function incorporated to the MSWord. In fact, Eqs. (1) and (2) are blurred.
The equations were described using the function incorporated to the MSWord.
10. In Table 3, the values of Young’s modulus are listed as those of “Compressive modulus”. However, there is a concern that these values are not confidential.
When reading the subsection 2.9, there are no descriptions how to measure the strain during the compression test. If the strain was measured from the crosshead movement, it could not be obtained accurately because of the take up of play in the testing machine and eccentricity in loading.
At least, the descriptions on the method how to measure the strain must be denoted in the subsection 2.9. If the authors think that the compression testing data are confidential enough, they must denote the issue clearly in the manuscript.
Since the elastic modulus is the slope of the tangent at the origin of the stress/strain curve, the tensile or compression modulus is called Young's modulus [Michel Biron. Chapter 3 - Basic criteria for the selection of thermosets. Thermosets and Composites. Technical Information for Plastics Users, 2004, PP. 145-181. https://doi.org/10.1016/B978-185617411-4/50005-X]. So in our manuscript, the Young's modulus term was used to explain the compressive modulus.
Please take into your consideration that the descriptions on the method how to measure the strain were added to the manuscript.
11. The presentation of Table 6 must be elaborated.
Please, be informed that the presentation of Table 6 was changed.
12. It is difficult to understand why the authors use a pair of photographs for Fig. 7. I think that either photograph is not necessary. This issue is also applicable to Fig. 8.
The pair of photographs in Figure 7 and Figure 8 was presented for better visualization of each layer of the cylindrical multi- UHMWPE forms samples.
13. The caption of Fig. 9 is too terse. It is necessary to demonstrate what the authors would like to represent by using the left and right photographs in the caption. This issue is also applicable to Fig. 10.
Please, be informed that the caption of Figure 9 was changed as follows:
“Figure 9. Cylindrical multi- UHMWPE forms samples, made of porous UHMWPE as trabecular layer and armored layer (cortical layer) that consists of bulk and films of UHMWPE. a) The sample appearance in the end of the mechanical test, b) dissected sample that shows the fixation of the glued multi- UHMWPE forms after the mechanical test.”
In addition, the Figure 10 was deleted.
14. The authors insist on the feasibility of the materials for the replacement of bone defects, etc., because the Young’s modulus value of the materials is close to that of bone. However, this issue is dubious at all because of the irrelevancies in measuring the Young’s modulus. Additionally, there are no descriptions on the Young’s modulus values of bones; therefore, it is difficult to evaluate the quality of the materials fabricated in this study.
Please, check the reformulated paragraphs:
“Since the obtained mechanical results revealed the good properties of the multi-UHMWPE forms composites (Figure 7 and Figure 8) glued by some glue types based on thiol-ene reaction, PVA/Cellulose and phenol formaldehyde; and also since these materials can be considered biocompatible; they can be considered as a promising development for joint reconstruction, such as implants for the replacement of bone defects or screws for osteofixation and orthognathic surgery. In addition, the Young’s modulus value (compressive modulus) of these composites is close to the one of the natural bone, which is in range of 6-30 GPa [45, 46], in comparison with the commercially available products based on metals, which has very high modulus value as presented in the introduction section [4]. This will lead to a successful bone remodeling and prevent bone resorption. Of course, the obtained compression mechanical properties of the prepared multi- UHMWPE forms composites do not fully match ones of the natural bone, but the proposed approach of using highly strengthened UHMWPE films as a reinforcement for the implants based only on UHMWPE without metallic reinforcement can be considered a promising approach.”
“The challenging and important problems for future studies are the enhancement of the glued fixation and the improvement of the mechanical properties of the prepared composites using the proposed technique that depends on the oriented UHMWPE films as a reinforcement material in order to fully match the mechanical properties of the natural bone.”